# Chitosan Sponges for Efficient Accumulation and Controlled Release of C-Phycocyanin

**DOI:** 10.3390/biotech12030055

**Published:** 2023-08-17

**Authors:** Yana E. Sergeeva, Anastasia A. Zakharevich, Daniil V. Sukhinov, Alexandra I. Koshkalda, Mariya V. Kryukova, Sergey N. Malakhov, Christina G. Antipova, Olga I. Klein, Pavel M. Gotovtsev, Timofei E. Grigoriev

**Affiliations:** 1Department of Biotechnology and Bioenergy, National Research Centre “Kurchatov Institute”, 123182 Moscow, Russia; yanaes2005@gmail.com (Y.E.S.); suhinov.dv@phystech.edu (D.V.S.); mar-1-ya@yandex.ru (M.V.K.); gotovtsevpm@gmail.com (P.M.G.); 2Department of NBIC-Technologies, Moscow Institute of Physics and Technology, National Research University, 141701 Dolgoprudny, Moscow Region, Russia; 3Department of Nanobiomaterials and Structures, National Research Centre “Kurchatov Institute”, 123182 Moscow, Russia; bestiamalum@yandex.ru (A.A.Z.); antipova_kg@nrcki.ru (C.G.A.); klein_olga@list.ru (O.I.K.); 4Faculty of Biotechnology, Mendeleev University of Chemical Technology of Russia, 125047 Moscow, Russia; a.koshkalda@bk.ru; 5Department for Resource Centre, National Research Centre “Kurchatov Institute”, 123182 Moscow, Russia; malakhov_sn@nrcki.ru; 6Research Center of Biotechnology of the Russian Academy of Sciences, A.N. Bach Institute of Biochemistry, 119071 Moscow, Russia

**Keywords:** C-phycocyanin, chitosan, sponges, *Arthrospira platensis*, antioxidant activity

## Abstract

The paper proposed a new porous material for wound healing based on chitosan and C-phycocyanin (C-PC). In this work, C-PC was extracted from the cyanobacteria *Arthrospira platensis* biomass and purified through ammonium sulfate precipitation. The obtained C-PC with a purity index (PI) of 3.36 ± 0.24 was loaded into a chitosan sponge from aqueous solutions of various concentrations (250, 500, and 1000 mg/L). According to the FTIR study, chitosan did not form new bonds with C-PC, but acted as a carrier. The encapsulation efficiency value exceeded 90%, and the maximum loading capacity was 172.67 ± 0.47 mg/g. The release of C-PC from the polymer matrix into the saline medium was estimated, and it was found 50% of C-PC was released in the first hour and the maximum concentration was reached in 5–7 h after the sponge immersion. The PI of the released C-PC was 3.79 and 4.43 depending on the concentration of the initial solution.

## 1. Introduction

Phototrophic microorganisms are a promising material for the environmentally sustainable production of a wide range of high value-added compounds (proteins, carbohydrates, lipids, and pigments) through the efficient use of artificial and actual sunlight. Among commercially used microalgae species, the cyanobacterium *p. Arthrospira* (commercially known as *Spirulina*) has been cultivated on a large scale for over 40 years as the basis of nutritional supplements and animal feed due to its high nutrient content and easy digestibility [1].

In addition, *Arthrospira* is a commercial source of C-phycocyanin (C-PC), a blue pigment–protein complex that has been approved by the Food and Drug Administration (FDA) as a natural blue food dye.

C-PC is an auxiliary pigment–protein complex involved in photosynthesis. In cyanobacteria, C-PC (λ_max_ 620 nm), along with C-phycoerythrin (λ_max_ 490–570 nm) and allophycocyanin (λ_max_ 650 nm), is part of phycobilisomes, a subcellular pigment–protein complex that optimizes the capture of light energy and provides its unidirectional highly efficient transfer to the reaction center of a photosystem II [2]. C-PC consists of α- and β-subunits (ratio 1:1, molecular weight 16–17 kDa and 18–19 kDa, respectively), to which one or two phycocyanobilins, linear tetrapyrrole open-chain chromophores, are attached via thioether bonds at cysteine residues [3]. The purity index (PI), which is determined by the ratio of absorption maxima at wavelengths of 620 nm and 280 nm (A_620_/A_280_), determines the area of C-PC use: the purity index is 0.5–1.5 for food-grade phycocyanin; 1.50–2.50 for cosmetic dye; 2.5–3.5 for biomarkers; not less than 3.5 for therapeutic agents (biomedical application); and more than 4.0 for analytical application [4]. Moreover, C-PC demonstrates antioxidant, neuroprotective, anti-inflammatory, and hepatoprotective properties [1].

Results of recent studies on C-PC mechanisms’ action on human skin are considered in the review by Dranseikien et al. [5]. It should be noted that the effect of C-PC on the wound-healing process was shown for the first time by Madhyastha et al. [6], in which it was found C-PC enhances fibroblast proliferation, whereas the maximum efficiency was observed in the concentration range from 50 to 100 µg/mL, but at 200 µg/mL, a cytotoxic effect was observed. The dose-dependent effect of C-PC on fibrinolytic activity was described in [7]: when cells were incubated in the presence of a C-PC solution with a concentration of 5 μg/mL (PI 3.8), the area of fibrin lysis increased by 30%, and at a concentration of 200 μg/mL, by 250%. In addition, C-PC promotes increased collagen synthesis [8], which accelerates the process of wound closure and also increases the expression of TGF-α1 and IL-1β factors, which are involved in the growth of fibroblasts and keratinocytes [9]. The presence of C-PC accelerates cell growth by inducing the G1 phase and maintaining proliferation without affecting the S and G2 phases [5].

The wound-healing activity of phycocyanin is based on its diverse properties, including its high antioxidant activity [9]. In addition, C-PC is able to fluoresce, which can be used to create systems with the ability to monitor wound healing.

A number of studies have also revealed the antibacterial effect of C-PC on various microorganisms, including the Gram-negative bacteria *Escherichia coli* and *Klebsiella pneumonia* [10] and Gram-positive bacteria *Propionibacterium acne* and *Staphylococcus epidermidis*, the increased number of which can lead to inflammatory processes of the skin [11]. In the work of Murugan and Radhamadhavan [12], it was shown that C-PC with a PI of 1.17 at concentrations of 40 µg/mL and 80 µg/mL did not have any effect on the growth of pathogenic fungi (*Candida albicans*, *Aspergillus niger*, *Aspergillus flavus*, and also representatives of the river *Penicillium* and *Rhizopus*), whereas C-PC with a PI of 3.74 inhibited fungal growth at a concentration as low as 45 µg/mL.

Studies have been conducted on the incorporation of C-PC into polymer carriers (hydrogel [8], microcapsules [13], liposomes [14]), the kinetics of C-PC release from the polymer matrix, and the subsequent wound-healing effect in experiments in vitro and in vivo. Thus, C-PC is a promising component of wound-healing materials, and its inclusion in the polymer matrix allows one to control the rate of its release, prolonging its action.

Chitin is a natural compound that composes the basis of the exoskeleton of many invertebrates: crustaceans, insects, and annelids [15]. In organisms, chitin forms complexes with proteins, lipids, pigments, and calcium carbonate. In addition, chitin is found in fungi (*Mucor rouxii* and *Choanephora cucurbitarum*) and algae (some diatoms). Unlike animal chitin, fungal chitin is more homogeneous and does not contain inorganic salts; however, it is linked to other polysaccharides, such as cellulose, glucan, and mannan, which makes it difficult to isolate [16].

Chitosan is an aminopolysaccharide composed of two types of β(1–4) linked structural units, namely 2-amino-2-deoxy-d-glucose and N-acetyl-2-amino-2-deoxy-d-glucose (Figure 1). This polymer is obtained by alkaline treatment of chitin, during which the polymer units undergo deacetylation. As a result, primary amino groups appear in the chitosan chain, capable of protonation in an acidic medium, the presence of which determines the solubility of chitosan in solutions of certain acids [17].

In addition, the presence of primary amino groups is the basis for most of the biological effects of chitosan, namely, antimicrobial, anti-inflammatory, and fungicidal action, as well as hemostatic and wound healing ability.

The wound-healing effect of chitosan oligomers is associated with their ability to stimulate the production of a number of enzymes—lysozyme, chitinase, and N-acetyl-b-D-glucosaminidase—the hydrolytic effect of which promotes macrophage activation and collagen synthesis [18]. Wound healing based on the activation of macrophages by chitosan is associated with the presence of receptors for GlcNAc-glycoproteins in the latter.

The use of a chitosan-based sponge impregnated with fibroblast growth factors (FGF) has been described in [19]. The inclusion of growth factors in sponges made of chitosan ensures a uniform distribution of the active substance and minimizes local irritation during drug administration [20].

A positive effect of various active substances loaded into the matrix of chitosan is shown in the process of healing diabetic wounds [21,22].

Upon contact with physiological fluids, the chitosan sponge forms a hydrogel, thus ensuring the delivery of the active substance even to deep wounds, unlike gels or ointments. As the sponge is destroyed under the action of enzymes, a steady release of the active substance is carried out. In addition, removal of the sponge after the wound has healed is most often not required.

The ability of chitosan to stabilize phycocyanin, preventing its thermal degradation has also been reported [23,24].

The aim of this work is to prepare chitosan sponges and study in vitro accumulation of C-PC isolated from the biomass of the cyanobacterium *Arthrospira platensis* by them, as well as the C-PC’s release properties and its characteristics after release.

## 2. Materials and Methods

### 2.1. Strain and Cultivation Conditions

C-phycocyanin was obtained by exhaustive extraction from the biomass of cyanobacterium *Arthrospira platensis* B-12619 (Russian National Collection of Industrial Microorganisms). *A. platensis* was grown in modified Zarrouk medium [25] consisting of (g/L): NaHCO_3_, 13.61; Na_2_CO_3_, 4.03; KH_2_PO_4_, 0.5; NaNO_3_, 2.5; K_2_SO_4_, 1.0; NaCl, 1.0; MgSO_4_·7H_2_O, 0.2; CaCl_2_, 0.03; FeSO_4_·7H_2_O, 0.01; and Na_2_-EDTA·2H_2_O, 0.08 in a batch mode. The initial pH was 9.25 ± 0.05. The cultivation was performed in 500 mL Erlenmeyer flasks, containing 200 mL of the Zarrouk medium. Flasks were kept in shaker-incubator Innova 42R (Eppendorf New Brunswick) set at 140 rpm under 30 °C, continuous lighting, and PAR of 16 ± 2 μmol/(m^2^·s). The optical density of the cyanobacteria suspension at 750 nm (OD_750_) was used to control biomass growth. The initial OD_750_ was 0.05 ± 0.01. The cyanobacteria biomass was harvested at the end of the exponential growth phase by centrifugation at 12,300× *g* for 15 min at 25 °C (Thermo Scientific SL40R, Thermo Fisher Scientific, Waltham, MA, USA). The biomass was washed twice with distilled water and re-centrifuged to remove adsorbed salts [26,27] and stored at −20 °C.

### 2.2. C-Phycocyanin Extraction and Purification

For exhaustive C-PC extraction, the *A. platensis* wet biomass was subjected to three cycles of freezing/thawing, dissolved in 0.2 M sodium phosphate buffer (pH 7.00 ± 0.02), and left incubating at 4 °C in the dark overnight [28]. Next, the samples were centrifuged (17,300× *g*, 30 min, 4 °C) to separate the supernatant, which was further purified and used in all experiments.

The C-PC crude extracts were purified by ammonium sulfate precipitation. At each purification step, the C-PC concentration and purity were controlled. Ammonium sulfate was gradually added to the crude extracts to achieve 20, 40, and 60% saturation with continuous stirring. The resulting solutions were kept overnight at 4 °C and centrifuged (17,300× *g*, 30 min, 4 °C). The obtained precipitate was dissolved in 0.2M sodium phosphate buffer (pH 7.00 ± 0.02). At the final stage (60% ammonium sulfate saturation) the C-PC precipitate was dissolved in deionized water and dialyzed against 100 volumes of deionized water overnight in the dark at 4 °C. The mixture was freeze-dried.

### 2.3. C-Phycocyanin Quantification and Purity Determination

The C-PC concentration was calculated according to Equation (1) [29], taking the absorption spectra with a Genesys 10S UV–visible spectrophotometer (Thermo Scientific, USA).
(1)$$C\text{-}\mathrm{PC}\;\left(\frac{\mathrm{mg}}{\mathrm{mL}}\right)=\frac{\left(A_{620}-0.474\times A_{650}\right)}{5.34},$$
where $A_{\lambda }$ is the absorbance of the extract at λ nm.

The C-PC purity index (*PI*) was determined according to Equation (2) [30].
(2)$$PI=\frac{A_{620}}{A_{280}},$$
where $A_{\lambda }$ is the absorbance of the solution at λ nm.

Equivalent amounts of the protein fractions were analyzed by protein gel electrophoresis SDS-PAGE (13%). 

### 2.4. High-Performance Liquid Chromatography

HPLC was performed on an Agilent Technologies 1200 chromatograph equipped with a DAD detector on a Kromasil C_5_ (250 × 4.6 mm) column. The composition of the mobile phase was acetonitrile containing 0.1% trifluoroacetic acid (solution A) and 0.1% aqueous solution of trifluoroacetic acid (solution B). The flow rate was 1.2 mL/min. The solution ratio solution A/solution B was increased from 25% to 100% within 15 min. The volume of the injected sample was 50 μL. The column effluent was monitored using a photodiode array detector at 620 nm.

### 2.5. Antioxidant Activity Measurements

The DPPH free radical scavenging activity of the C-PC samples was evaluated according to [28]. The C-PC sample or antioxidant standard (Trolox) ethanol solution was added to the same volume of a 0.1 mM DPPH (Sigma-Aldrich, Burlington, NJ, USA) ethanol solution and incubated at 25 °C in the dark for 30 min. The absorbance was measured at 517 nm using ethanol as a blank. The percentage inhibition of free radicals was determined according to Equation (3).
(3)$$\mathrm{Inhibition}\;\left(\%\right)=\left(1-\frac{A_{\mathrm{sample}}}{A_{\mathrm{control}}}\right)\times 100$$
where A_sample_ is the absorbance of the treated sample, and A_control_ is the absorbance of 0.05 mM DPPH ethanol solution.

The antioxidant activity of the C-PC samples was expressed in terms of the effective concentration of IC_50_, i.e., the concentration required to inhibit 50% of the DPPH free radicals, which was calculated using the linear regression formula.

The Trolox equivalent antioxidant capacity (*TEAC*) was calculated according to Equation (4) using the DPPH free radicals inhibition curves for Trolox and the C-PC samples. The result was expressed in terms of micromoles of trolox equivalent (TE) per gram of the C-PC (µM TE/g):(4)$$TEAC=\frac{IC_{50}\;of\;Trolox\;\left(\mu \mathrm{mol}/L\right)}{IC_{50}\;of\;C\text{-}\mathrm{PC}\;\left(g/L\right)}$$

### 2.6. Chitosan Sponge Preparation

Sponges were prepared from Primex ChitoClear^®^ HQG 1600 (Iceland). Then, 200–300 μL of a 25% sodium hydroxide solution was added dropwise to a 2% (wt.) solution of chitosan in 2% (wt.) acetic acid with vigorous stirring to reach pH 5.7, and then 1% (wt.) of glutaraldehyde (Russia) was added (calculated on the dry weight of chitosan). The resulting solution was poured into a plate and frozen, after which it was freeze-dried on Martin Crist ALPHA 1-4LSC (Germany).

### 2.7. Incorporation of C-PC into a Chitosan Sponge, Encapsulation Efficiency (Entry Efficiency), Load Capacity, and In Vitro Release Study

C-PC was introduced into the chitosan sponge matrix by immersing the sponge (29.21 ± 2.24 mg) in 5 mL of an aqueous solution of C-PC at three test concentrations of 250, 500, and 1000 µg/mL (hereinafter, CPC-250, CPC-500, and CPC-1000) for 24 h at room temperature in the dark.

Next, the sponges were freeze-dried and stored at 4 °C until testing.

The relative concentration of C-PC ($C_{r}$, %) is the remaining concentration of C-PC ($C_{t}$) as a percentage of the initial concentration ($C_{0}$) according to the equation:(5)$$C_{r}\;\left(\%\right)=\frac{C_{t}}{C_{0}}\times 100,$$

The amount of C-PC transferred to the sponge was determined by the formula:(6)$$m_{C\text{-}\mathrm{PC}}=\left(C_{0}\times V_{0}\right)-\left(C_{t}\times V_{t}\right),$$
where $C_{0}$, $V_{0}$ and $C_{t}$, $V_{t}$ are concentrations and volumes of C-PC solution at the initial and final time, respectively.

The entry efficiency (*EE*) was calculated by the formula: (7)$$EE\;\left(\%\right)=\frac{m_{0}-m_{un}}{m_{0}}\times 100,$$
where $m_{0}$ and $m_{un}$ are the initial mass and mass of unloaded C-PC, respectively.

Load capacity (*LC*):(8)$$LC\;\left(mg/g\right)=\frac{m_{L}}{m_{S}}\times 100,$$
where $m_{L}$ (mg) and $m_{S}$ (g) are the mass of the loaded C-PC and the mass of the sponge, respectively.

To study the release of loaded C-PC, C-PC loaded sponge samples were placed in 10 mL of saline (pH 7.4) and kept at 37 °C and 100 rpm for 24 h. 

### 2.8. Microscopy Imaging 

Microscopy images were obtained by using a fluorescence microscope BX61 (Olympus, Tokyo, Japan) with a DP71 digital microscope camera (Olympus, Japan).

Sponges (1 mm thick) were incubated for 20 min in a support medium (OCT, Tissue-Tek; Sakura Finetek USA, Torrance, CA, USA) and then cooled to −30 °C. Cryosections (20 μm thickness) were prepared using Cryostat Microm HM 525 (Thermo Scientific). 

### 2.9. Fourier Transform Infrared Spectroscopy Measurements (FTIR)

IR spectra were recorded using a Nicolet iS5 IR Fourier spectrometer (Thermo Fisher Scientific, USA) with an iD5 ATR frustrated total internal reflection attachment (crystal–diamond). The spectral range was 4000–550 cm^−1^, the spectral resolution was 4 cm^−1^, the number of scans was 32. The spectra were recorded and processed using the standard software of the instrument (Omnic 8.2).

### 2.10. Scanning Electron Microscopy of Sponges (SEM)

SEM images of the chitosan-based sponge were obtained using a scanning electron microscope Thermo Scientific Phenom XL with a backscattered electron detector at an accelerating voltage of 5 kV and a pressure of 60 Pa without a conductive coating.

### 2.11. Statistical Analysis

Numerical results are presented as means ± SD of at least three independent replicates. One-way analysis of variance (ANOVA) followed by Tukey’s test were used to verify significant differences considering a confidence level of 95% (*p* < 0.05).

## 3. Results and Discussion

At the moment, the market price of C-PC with a purity of more than 3.5 exceeds 150EUR per 1 mg [31], and commercial products containing C-PC of similar purity will have a high cost. Therefore, in our studies, we used C-PC with a PI of 3.36 ± 0.24, obtained using a simple and cheap process for extracting C-PC from the wet biomass of the cyanobacterium *A. platensis*.

Figure 2 shows optical and fluorescent microscopy of the cyanobacterium *A. platensis*, as well as the absorption and fluorescence spectra of the C-PC isolated from it, which was used for the experiments with sponges.

### 3.1. Extraction and Purification of Phycocyanin

C-PC was extracted from the wet biomass and isolated by 3 cycles of freeze/thawing in phosphate buffer (pH 7.0). As a result, a crude extract was obtained, the absorption spectrum of which had two main maxima in the region of 280 nm (protein part) and 620 nm (chromophore group), as well as a small shoulder at 650 nm (Figure 3), indicating the presence of trace amounts of allophycocyanin (APC), the PI was 1.53. Purification of the C-PC was carried out by stepwise precipitation with ammonium sulfate. For this, 20% ammonium sulfate was initially added to the crude extract, the precipitate was discarded, and 40% ammonium sulfate was added to the supernatant solution. At this stage, the separation of the C-PC that precipitated and the APC that remained in the solution occurred, as evidenced by the absence of a shoulder at 650 nm in the absorption spectrum. Next, the saturation with ammonium sulfate was increased to 60%, which led to the complete precipitation of the C-PC. The precipitated pigment was dissolved in milliQ, dialyzed overnight against water, and then lyophilized. The resulting powder had a PI of 3.36 ± 0.24. Prior to testing, C-PC powder was stored at 4 °C.

During the purification of the crude C-PC, the change in the PI, along with the absorption spectra, was also controlled by gel electrophoresis. Figure 3b clearly shows two bands at about 17 kDa and 19 kDa, corresponding to the α- and β-subunits of C-PC. In addition, these pictures show that with each subsequent stage of purification, the PI of the C-PC increased, the intensity of the absorption maximum at 280 nm decreased (Figure 3a), while the number of bands corresponding to impurity proteins (molecular weight different from the mass of α- and β-subunits) also decreased (Figure 3b).

### 3.2. Characteristics of Pure C-PC

Also, pure C-PC was further characterized by reverse phase HPLC (Figure 4), FTIR, and its AOA was also determined.

Detection at 620 nm revealed two major peaks (7.66 and 8.02 min) which were identified as the α- and β-subunits of C-PC. The β-subunit contains twice as much phycocyanobilin chromophore as the α-subunit, so the peak attributed to the β-subunit is more intense. The results obtained are consistent with the data from the literature [32,33,34].

The pure C-PC was also examined by FTIR (Figure 5). The IR spectrum of the C-PC contains specific bands of amide I (C=O stretching vibrations) and amide II (N–H bending + C–N stretching) at 1650 cm^−1^ and 1536 cm^−1^, respectively; and the intensity of the amide I band is higher than amide II. The position and shape of the amide I band are used to analyze the secondary structure of the protein. The sharp peak of the amide I band points to the α-helix as the main element of its secondary structure [11]. It should be noted that the IR spectrum of the lyophilized pure C-PC additionally confirmed the absence of impurities of inorganic sulfates and phosphates (absence of intense bands in the region of 1040 and 1015 cm^−1^).

Antioxidant activity was determined by the radical scavenging activity of DPPH (2,2-diphenylpicrylhydrazyl). For the pure C-PC, the IC_50_ was 212.73 µg/mL, 174.47 *TEAC*.

### 3.3. Chitosan Sponges

Based on chitosan Primex ChitoClear^®^ HQG 1600, sponges were obtained by lyophilization of an acidic aqueous solution. The standard method involves freeze-drying an aqueous chitosan solution in acetic acid. However, in this work, the approach was slightly modified. An important factor affecting the stability of phycocyanin is the acidity of the medium. According to the data from the literature [35], C-phycocyanin retains its stability at medium with pH in the range of 5.5–6.0. At higher or lower values, degradation of C-phycocyanin occurs with a change in its spectral properties and color.

A chitosan sponge placed in an aqueous solution causes a significant decrease in the pH of the medium by washing out the acetic acid remaining in the sponge after lyophilization. In order to combine the sponge with the phycocyanin solution, before lyophilization, the pH of the aqueous acid solution of chitosan was adjusted to a value of 5.7 by adding a small amount (200–300 µL) of a 25% aqueous solution of sodium hydroxide. This pH value approximately corresponds to the complete neutralization of free protons not bound to the amino groups of chitosan (Figure 6). At the same time, the polymer does not precipitate due to intensive mixing. The use of a concentrated alkali solution makes it possible to avoid dilution of the chitosan solution and, as a result, a decrease in the mechanical characteristics of the sponges.

When the first equivalence point is reached, the free acid is neutralized. The first equivalence point roughly corresponds to the solution pH of 5.6. Further titration is based on the interaction of the titrant with the protonated amino groups of chitosan.

Since the pore wall thickness of chitosan sponges is quite small, the pore wall thickness does not exceed 100 μm (Figure 7d), and most of the protonated amino groups of chitosan are probably located on the wall surface. At pH 5.7, phycocyanin, whose isoelectric point corresponds to pH 4.6, is predominantly negatively charged, which contributes to more efficient adsorption of the protein complex on the surface of the chitosan sponge due to the forces of electrostatic interaction.

### 3.4. Sponge Chitosan–Phycocyanin

With the introduction of C-PC into the matrix of the chitosan sponge, three concentrations were tested: 250, 500, and 1000 µg/mL (hereinafter, CPC-250, CPC-500, and CPC-1000). The choice of tested concentrations was based on the analysis of the data from the literature [6,7,8,9,36,37,38,39,40,41]. As can be seen in Figure 8, at the lowest concentration tested (CPC-250), half of the initial amount of C-PC passed into the sponge in about 2.5 h, while the other two concentrations took longer, 4 and 5 h, respectively, for CPC-500 and CPC-1000. After 8 h from the beginning of the experiment, 72.35 ± 2.39, 64.64 ± 1.26 and 57.05 ± 0.73% of the initial amount of C-PC passed into the sponges for CPC-250, CPC-500, and CPC-1000, respectively. Over the next 16 h, the amount of C-PC that did not load into the sponge (remaining in the solution) was 18.90, 20.20, and 24.25%, respectively.

Based on the results obtained, the entry efficiency (*EE*) and load capacity (*LC*) were calculated (Table 1).

As can be seen from the table, the entry efficiency (*EE*), regardless of the initial concentration of C-PC, was more than 90%, which exceeded the available literature data on loading C-PC into polymer carriers: 67% and 72% with STMP/STPP C-PC encapsulation cross-linked starches from different botanical sources [29] and phycocyanin–alginate beads [42], respectively.

After the introduction of C-PC into the sponge and subsequent drying, the sponges acquired a blue tint, the intensity of which depended on the concentration of C-PC tested. Optical and fluorescence microscopy of thin sections of the sponges was carried out for the sponges loaded with C-PC (Figure 9), and the FTIR spectrum was also taken (Figure 5).

Figure 5 shows the spectra of both the initial C-PC, the chitosan sponge, and the sponge loaded with C-PC. Chitosan contains a large number of amino groups in its composition, as a result, amide I and II bands are also present in its spectrum, and the intensity of the amide I band is significantly lower than amide II. As a consequence, an increase in the intensity of the peak at the amide I band in the spectrum of the loaded sponge compared to the spectrum of the original sponge indicates the inclusion of C-PC in the polymer matrix, and the absence of new peaks indicates the absence of the formation of new bonds.

### 3.5. C-PC Release from the Sponge

In this study, the release of C-PC from the sponges into saline (pH 7.4) was studied. The in vitro release profiles of C-PC from chitosan sponges are shown in Figure 10.

At the lowest of the tested concentrations (CPC-250), the maximum concentration of C-PC released from the sponge into the solution was reached in 5 h after the sponge was immersed in saline and remained practically unchanged until the end of the experiment, while more than 50% of the initial amount remained in the sponge.

For the other two concentrations, the release rate was faster: 50% of C-PC from the sponges obtained with CPC-500 was released after 1 h, and it took less than 1 h for CPC-1000. The maximum amount of C-PC came out at 7 h, and the residual content of C-PC in sponges after 24 h was about 20% and 15%, respectively.

For C-PC, released from the sponge, the PI and AOA were determined (Table 2).

According to the results, for all variants of the experiment, the PI of the C-PC released from the sponge was higher (therapeutic and analytical purity) than that of the initial C-PC. In addition, the AOA was also higher than the initial value (by 32, 45, and 82%, respectively, for CPC-250, CPC-500, and CPC-1000), and there is a clear dependence of the increase in the AOA with increasing the PI.

## 4. Conclusions

In the work, sponges based on chitosan containing C-PC were obtained. The encapsulation efficiency value exceeded 90%, and the maximum loading capacity was 172.67 ± 0.47 mg/g. FTIR analysis showed that no new bonds are formed during the introduction of C-PC into the sponge. An in vitro study showed that about 50% of C-PC was released in the first hour, and the released C-PC reached a maximum concentration at 5 h, which remained at a constant level during the next 19 h of control. The released C-PC was characterized by a PI consistent with pharmaceutical (3.79) and analytical (4.43) uses and an increased AOA.

Thus, as a result of the studies carried out, a C-PC-containing sponge was obtained, which can be used as an integumentary material for wound healing.

## Figures and Tables

**Figure 1 biotech-12-00055-f001:**
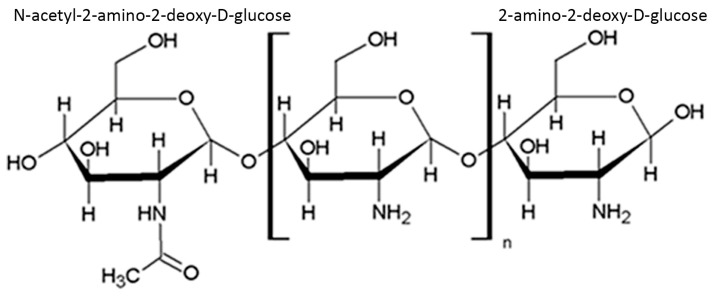
Structure of chitosan.

**Figure 2 biotech-12-00055-f002:**
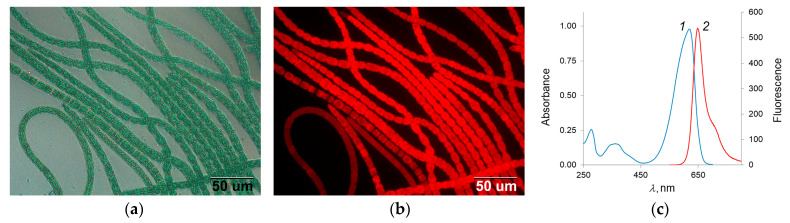
Microscopy of the cyanobacterium *A. platensis*: (**a**) optical; (**b**) fluorescent; (**c**) C-PC spectra: absorption (1) and fluorescence (2).

**Figure 3 biotech-12-00055-f003:**
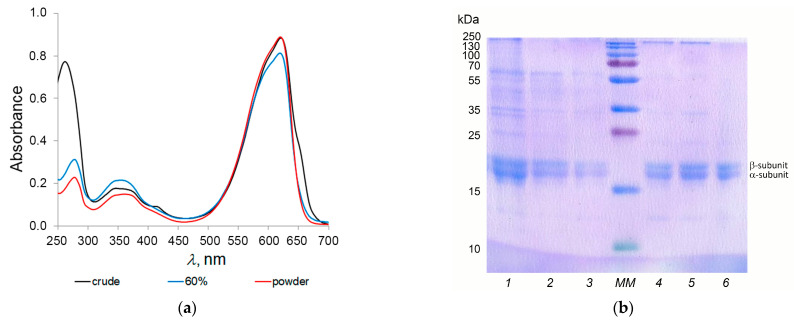
Control of changes in the PI of C-PC: (**a**) absorption spectra; (**b**) SDS-PAGE: 1—crude, ammonium sulfate precipitation: 2–20%; 3–40%; 4–60%; 5—after dialysis; 6—powder; MM—molecular markers.

**Figure 4 biotech-12-00055-f004:**
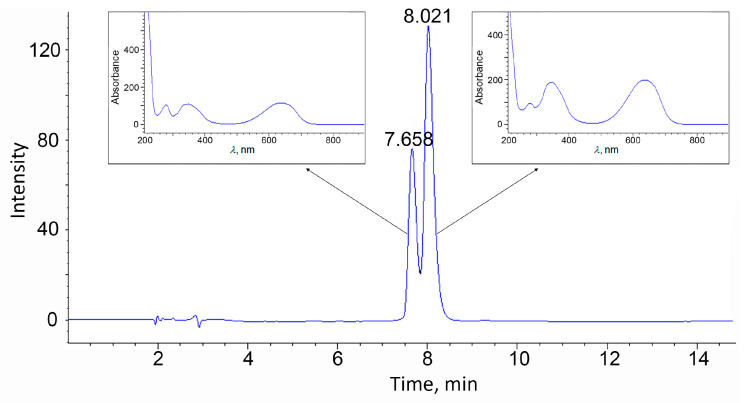
HPLC of pure C-PC and absorption spectra of the corresponding peaks.

**Figure 5 biotech-12-00055-f005:**
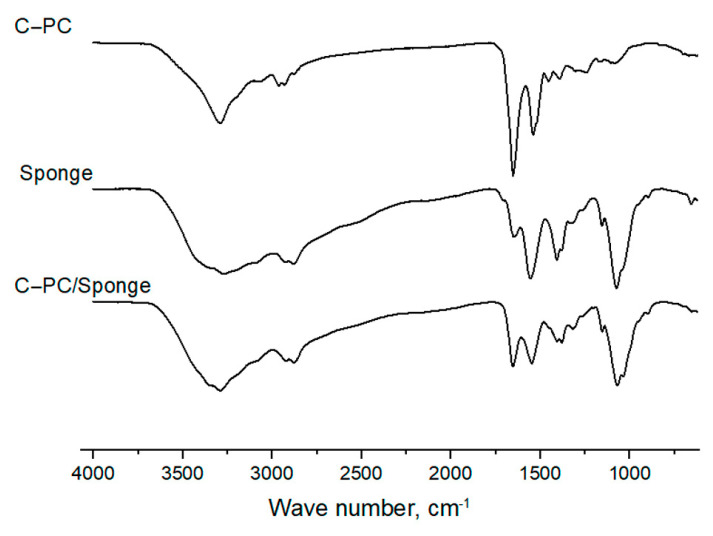
FTIR spectra of C-PC, chitosan sponge, and sponge with C-PC.

**Figure 6 biotech-12-00055-f006:**
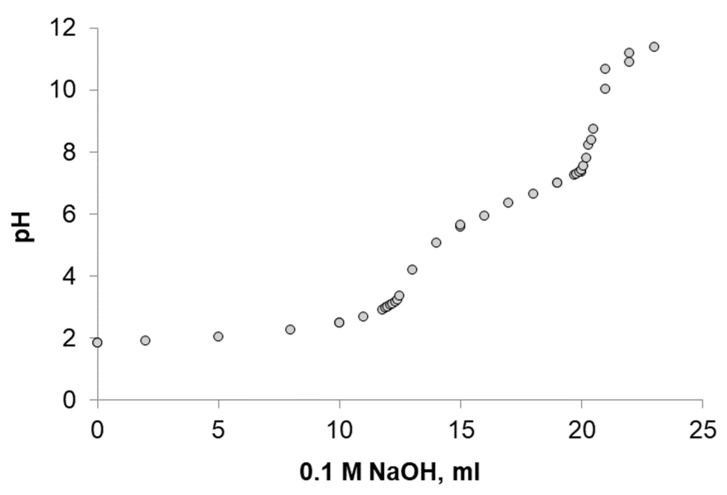
Titration curve of 1% chitosan solution in 0.1 M hydrochloric acid.

**Figure 7 biotech-12-00055-f007:**
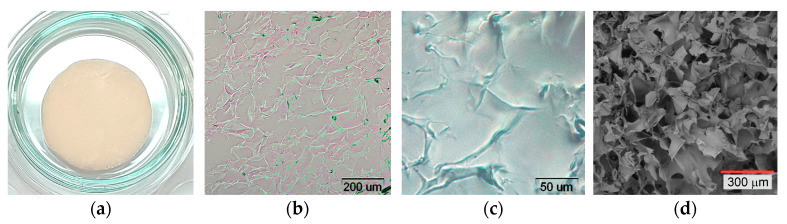
Chitosan sponges: (**a**) appearance; (**b**,**c**) thin layer microscopy; (**d**) SEM-image.

**Figure 8 biotech-12-00055-f008:**
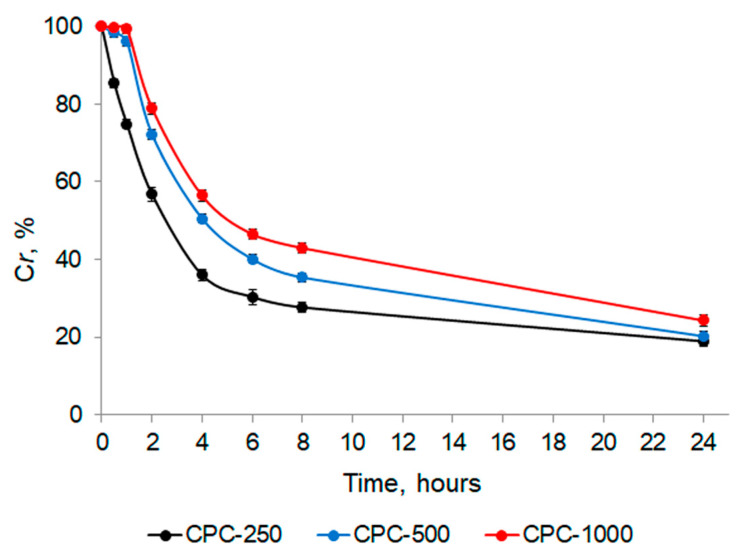
Introduction of C-PC into the sponge.

**Figure 9 biotech-12-00055-f009:**
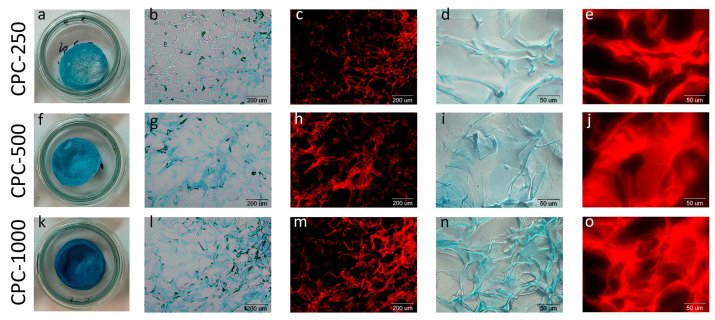
Appearance of the chitosan sponges (**a**,**f**,**k**) and thin section optical (**b**,**d**,**g**,**i**,**l**,**n**) and fluorescent (**c**,**e**,**h**,**j**,**m**,**o**) microscopy with magnification ×40 (**b**,**c**,**g**,**h**,**l**,**m**) and ×100 (**d**,**e**,**h**,**i**,**n**,**o**): CPC-250 (**a**–**e**), CPC-500 (**f**,**g**), CPC-1000 (**k**–**o**).

**Figure 10 biotech-12-00055-f010:**
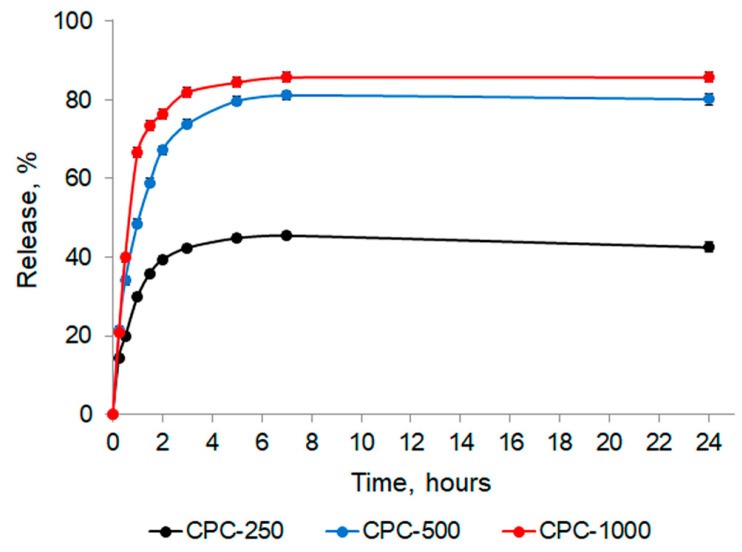
In vitro release profiles of C-PC from the chitosan sponges.

**Table 1 biotech-12-00055-t001:** Efficiency and loading capacity.

	CPC-250	CPC-500	CPC-1000
*EE*, %	95.86 ± 0.26	94.17 ± 0.55	92.84 ± 0.50
*LC*, mg/g	38.01 ± 2.02	82.03 ± 0.38	172.67 ± 0.47

**Table 2 biotech-12-00055-t002:** C-PC PI and antioxidant activity.

		C-PC	CPC-250	CPC-500	CPC-1000
PI		3.36 ± 0.24	3.55 ± 0.16	3.79 ± 0.08	4.43 ± 0.06
DPPH	IC_50_, µg/mL	212.73	161.47	146.12	116.81
	TEAE	174.47	229.85	253.99	317.73

## Data Availability

Not applicable.

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
