# Peer review of "Chitosan Sponges for Efficient Accumulation and Controlled Release of C-Phycocyanin"

_biotech, 2023, doi:10.3390/biotech12030055_

Round 1

Reviewer 1 Report

1. In Figure 2, the authors do not explain what the peak at 350 nm in the C-PC spectrum represents.

2. The formulae for calculating the residual content of C-PC in solution (Cr) and the residual content in the sponge (CR) on page 5 are incorrect. I hope the author will proofread them again.

3. Background descriptions for chitosan can be strengthened by citing 10.1016/j.cej.2023.141852; 10.1016/j.ijbiomac.2021.12.007 and what are the advantages of the current work compared to published articles?

4. The authors should have explained why the CPC concentrations of 250, 500, and 1000 ug/mL were chosen.

5. Figure 2b would be better if some processing could be done.

6. In Figure 10, why does the curve of CPC-250 slowly slope upward after 7 hours, which means that some of the CPC has returned to the sponge?

Author Response

Dear Reviewer,

we wish to thank you for their helpful criticisms, all of which we have took into account and considered in preparing our revised manuscript. We have adjusted the manuscript in accordance with your comments. We kindly ask you to find explanations below:

  1. In Figure 2, the authors do not explain what the peak at 350 nm in the C-PC spectrum represents.

The phycocyanabillin molecule is very flexible and can take on various conformations. In the native C-PC molecule, phycocyanobilin (chromophore) is linked to protein chains through thioether bonds. As a result, it assumes an extended conformation and the absorption maximum lies in the region of 615–620 nm. If the stabilizing bonds between the chromophore and the protein are lost, the chromophore rearranges itself into its energetically favorable cyclic conformation. Changes in the conformation of the chromophore lead to a different form of the p-electron system and, accordingly, to different absorption spectra. In cyclic conformation phycocyanobilin has a maximum absorption in the region of 350-360 nm.

Thus, the relative intensity of the two main bands is an indicator of the chromophore conformation: the more intense band at 350 nm corresponds to cyclic conformations of the “porphyrin type”, and the more intense band at 620 nm indicates stretching of the chromophore (linear conformation).

As mentioned above, in the native state of C-PC, the chromophore is in the linear conformation and that is why the ratio A620/A280 is usually used to assess the purity of C-PC (after extraction and during purification steps), which indicates the purity of C-PC compared to impurity (contaminating) proteins (absorption at 280 nm). And the invariance of the A360 intensity with respect to A620 indicates the preservation of the structure, i.e. isolation and purification conditions had no effect on the C-PC structure.

  1. The formulae for calculating the residual content of C-PC in solution (Cr) and the residual content in the sponge (CR) on page 5 are incorrect. I hope the author will proofread them again.

Corresponding changes have been made to the text of the article.

  1. Background descriptions for chitosan can be strengthened by citing 10.1016/j.cej.2023.141852; 10.1016/j.ijbiomac.2021.12.007 and what are the advantages of the current work compared to published articles?

The reference section was extended by adding these publications.

As advantages compared to published data on the encapsulation of C-PC in various matrices used for healing, it should be noted high loading capacity and encapsulation efficiency, high release %, as well as an increase in the purity of C-PC at the outlet of the sponge. Thus, the stage of chromatographic purification of C-PC to analytical purity is excluded from the general scheme of the process, which makes the process more cost-effective, since the C-PC purification step can be limited to only concentration and precipitation with ammonium sulfate.

  1. The authors should have explained why the CPC concentrations of 250, 500, and 1000 ug/mL were chosen.

Since the loading capacity of the studied chitosan sponges in relation to C-PC has not been studied, the choice of the test concentrations of C-PC was based on the literature data: on the non-toxicity of C-PC for fibroblasts (no more than 200 μg/ml) and on the stimulation of cell proliferation and migration concentrations above 25 µg/ml (https://doi.org/10.1007/978-3-030-75506-5_54); on increasing the proliferation of fibroblasts with the best effect at a concentration of C-PC 75 µg/ml (https://doi.org/10.5897/JMPR12.705 or https://academicjournals.org/journal/JMPR/article-abstract/DDDFE5F20765); on the effect of C-PC extracts on processes involved in human keratinocyte proliferation, regeneration and migration, where C-PC showed the best growth promotion effect at a dose of 33.5 µg/ml (https://doi.org/10.5897/JMPR12.705).

We also considered the data on the wound healing effect of C-PC in the form of a solution (DOI: 10.1055/s-0029-1234384, https://doi.org/10.1134/S0003683820050166), imbedded into a polymer matrix (https://doi.org/10.1016/j.scp.2022.100905), or as an ingredient in skin cream recipe (https://doi.org/10.1080/13880209.2017.1331249, https://doi.org/10.1007/s10811-023-02988-z).

The corresponding phrase about the prerequisites for choosing test concentrations has been added to the text of the article.

  1. Figure 2b would be better if some processing could be done.

The picture has been replaced with a clearer one.

  1. In Figure 10, why does the curve of CPC-250 slowly slope upward after 7 hours, which means that some of the CPC has returned to the sponge?

With the release of C-PC from sponges obtained using only the lowest concentration tested (CPC-250), we observed some fluctuation in the concentration of released C-PC, on the basis of which we assumed that some balance of C-PC concentration was established outside and inside the sponge. and this is the subject of our more detailed study.

In addition we have attached this response in form of a file for your convenience.

Kind regards,

On behalf of all co-authors,

Dr. Timofei Grigoriev,

National Research Center «Kurchatov Institute»,

Akademika Kurchatova pl.,1, Moscow, 123182, Russia

Tel.: +7 (499) 1969539

Fax: +7 (499) 1961704

E-mail address: [email protected] (Timofei E. Grigoriev)

Reviewer 2 Report

Manuscript Title : Chitosan sponges for efficient accumulation and controlled  release of C-phycocyanin

Manuscript ID : biotech-2519084.

The present study investigated the extraction of C-phycocyanin (C-PC) from cyanobacteria Arthrospira platensis biomass, then the use of C-PC loaded chitosan matrix as a porous material for wound healing.

The materials developed in this paper are very promising and the experimental work is important. The manuscript is well written structured. The language of the manuscript is good. Results found in this research work are of great interest.

The paper could be considered for publication in bioTech journal after authors address the comments  reported on the attached file : Review biotech-2519084.pdf

* Plagiarism total score is 28 %. Shall modify content taken from these two sources:

ü  8% from the work : Daniil V. Sukhinov, Kirill V. Gorin, Alexander O.; Romanov, Pavel M. Gotovtsev, Yana E. Sergeeva. "Increased C-phycocyanin extract purity by flocculation of Arthrospira platensis with chitosan", Algal Research, 2021.

ü  4% from Mdpi source.

Good English quality.

Author Response

Dear Reviewer,

we wish to thank you for their helpful criticisms, all of which we have took into account and considered in preparing our revised manuscript. We have adjusted the manuscript in accordance with your comments. We kindly ask you to find explanations below:

The article has been amended and supplemented in accordance with the attached file: Review biotech-2519084.pdf.

Concerning Plagiarism total score is 28 %:

Changes have been made to the text that referred to the article we published earlier (Daniil V. Sukhinov et al., Increased C-phycocyanin extract purity by flocculation of Arthrospira platensis with chitosan. Algal Research. 2021).

In addition, we would like to give some clarifications and explanations:

  1. 1. 2, lines 54-55. The text of the article provides a link to one source, from where the data was taken. In the general case, the accepted gradations of C-PC purity: food, cosmetic, therapeutic, analytical. For example, “product with PI greater than 0.7 is usually considered as food grade, while greater than 1.5 as cosmetic grade, and greater than 3.9 as reactive grade, and finally greater than 4.0 as analytical grade” [https://doi.org/10.1007/s11947-022-02926-w], “The phycocyanin purity index will determine its market value and, consequently, its applicability. Indeed, phycocyanin is considered food, cosmetic, reagent, or analytic grade when A620/A280 is, respectively, greater than 0.7, 1.5, 3.9, or 4.0” [https://doi.org/10.3390/ph16040592].

The referenced article provides even a finer division: ‘Furthermore, the extracts obtained also fit into the classification used by companies that sell C-PC, with purities between 0.5 to 1.5 for use as a food dye, between 1.50 and 2.50 for use as a cosmetic dye, between 2.5 and 3.5 for use as a biomarker, and a purity greater than 4 for use in biomedical applications and as a therapeutic agent” [dx.doi.org/10.1590/0104-6632.20180353s20170160]. Therefore, we considered it necessary to refer specifically to these data

  1. P. 2, line 67. Add an explaining scheme. This study is devoted to the study of the possibility of introducing C-PC into the polymer matrix (chitosan sponge) and its subsequent release from the matrix. A more detailed study of the effect of C-PC on the wound healing process is the subject of our current research, and we think it would be logical to present the scheme of C-PC participation in the Background of our next publication, so in this article we would like to leave the description in text form.

In addition we have attached this response in form of a file for your convenience.

Kind regards,

On behalf of all co-authors,

Dr. Timofei Grigoriev,

National Research Center «Kurchatov Institute»,

Akademika Kurchatova pl.,1, Moscow, 123182, Russia

Tel.: +7 (499) 1969539

Fax: +7 (499) 1961704

E-mail address: [email protected] (Timofei E. Grigoriev)

Round 2

Reviewer 2 Report

Manuscript Title : Chitosan sponges for efficient accumulation and                                              controlled  release of C-phycocyanin

Manuscript ID : biotech-2519084-V2.

The authors addressed the requested issues. They gave good explanations and added useful paragraphs, so the manuscript is more consistent and well written. The plagiarism rate is decreased to acceptable level.